# Epidemiological Impact of Increasing Vaccination Coverage Rate and Re-Vaccination on Pneumococcal Disease in Older Adults in Germany

**DOI:** 10.3390/vaccines13050475

**Published:** 2025-04-28

**Authors:** Oluwaseun Sharomi, Marion de Lepper, Sarah Mihm-Sippel, Thorsten Reuter, Claudia Solleder, Giulio Meleleo, Tufail M. Malik, Kevin M. Bakker, Rachel J. Oidtman

**Affiliations:** 1Merck & Co., Inc., Rahway, NJ 07065, USA; 2MSD Sharp & Dohme GmbH, 81673 Munich, Germany; 3Wolfram Research, Inc., Champaign, IL 61820, USA

**Keywords:** pneumococcal, IPD, vaccine, dynamic transmission model, uptake

## Abstract

Background/Objectives: The clinical impact of replacing the 23-valent pneumococcal polysaccharide vaccine (PPSV23) for the vaccination of older (≥60 years) and at-risk German adults with either the 20-valent (PCV20) or 21-valent (V116) pneumococcal conjugate vaccine (PCV) was evaluated. Methods: An age- and serotype-specific transmission model was adapted to Germany to evaluate the impact of V116 versus PCV20 vaccination on pneumococcal disease (PD) incidence, including invasive pneumococcal disease (IPD) and inpatient and outpatient non-bacteremic pneumococcal pneumonia, over 10 years. A reference strategy (PPSV23 vaccination at a constant 30% vaccine coverage rate (VCR)) was compared against eight strategies varying by PCV (PCV20 vs. V116), VCR (30% vs. 60%), with or without the PCV revaccination of previously PPSV23-vaccinated adults (0% vs. 50% revaccination). Results: Vaccination with PCV20 and V116 initially decreased PD incidence, but incidence returned to pre-vaccine levels after five and eight years, respectively. Increasing the VCR to 60% prevented this resurgence. At a 10-year time horizon, V116 with 30% VCR reduced IPD cases by 9%, inpatient NBPP cases by 10%, and outpatient NBPP cases by 7% compared to the reference strategy. PCV20 with 30% VCR reduced these cases by 6%, 5%, and 4%, respectively. Increasing the VCR to 60% and revaccinating 50% of previously PPSV23-vaccinated adults further reduced IPD cases by 14% and 13% for V116, and by 9% and 9% for PCV20. Conclusions: Increasing the vaccination coverage rate to 60% and strategically revaccinating previously PPSV23-vaccinated adults significantly enhanced the effectiveness of pneumococcal vaccines, with V116 showing greater overall reductions in disease incidence compared to PCV20 or PPSV23.

## 1. Introduction

Pneumococcal diseases, caused by the bacterium *Streptococcus pneumoniae*, remain a significant public health concern in Germany, particularly affecting young children, the elderly, and individuals with underlying health conditions [1]. Approximately 100 different *S. pneumoniae* serotypes have been identified, and several vaccine formulations, including pneumococcal conjugate vaccines (PCVs) and pneumococcal polysaccharide vaccines (PPSVs), are available which target varying subsets of these serotypes [1,2].

The history of pneumococcal vaccines in Germany dates back to the early 20th century. Initial efforts involved whole-cell vaccines, which were eventually replaced by polysaccharide vaccines. The first pneumococcal polysaccharide vaccine (PPSV14) was licensed in 1977, covering 14 serotypes. This was followed by the 23-valent pneumococcal polysaccharide vaccine (PPSV23) in 1983, which expanded coverage to 23 serotypes. However, these vaccines were less effective in young children and immunocompromised individuals [2,3].

The Standing Committee on Vaccination (STIKO) at the Robert Koch Institute recommended the introduction of PCV7 into the pediatric national immunization program (NIP) of Germany in July 2006 [4]. STIKO has since continuously updated its recommendations to include PCV10, PCV13, and PCV15 for the routine immunization of infants and young children. In 2025, PCV13 and PCV15 are recommended for routine PCV vaccination below 2 years of age in a 2 + 1 schedule for full-term infants and a 3 + 1 schedule for preterm infants. The pediatric vaccine coverage rate (VCR) in Germany for children up to 2 years of age who have completed the vaccination schedule was 74% as of 2023 [5]. For adults aged 60 years and older, STIKO has historically recommended the 23-valent pneumococcal polysaccharide vaccine (PPSV23) since 1998. Since September 2023, only PCV20 is recommended for adult vaccination [6]. Adult VCR remains low, with only 23.3% of adults aged 60–74 years having received the recommended pneumococcal vaccination as of 2022 [5].

The incidence of pneumococcal diseases (PD) in adults in Germany has shown notable trends over the years. From 2003 to 2006, prior to the inclusion of PCV7 in the NIP, the incidence of invasive pneumococcal disease (IPD) in adults aged 60 and older was 1.64 cases per 100,000 population. This incidence increased to 10.08 cases per 100,000 population in subsequent years [7]. From 2016 to 2019, the overall incidence rate of all-cause pneumonia (ACP) among individuals aged 16 years and older in Germany was 1345 per 100,000 person–years, while the incidence rate for IPD was 8.25 per 100,000 person–years [1]. These rates highlight the significant burden of PD among adults, particularly in older age groups and those with underlying health conditions [1]. Non-bacteremic pneumococcal pneumonia (NBPP), a form of pneumonia caused by *S. pneumoniae* without the presence of bacteria in the bloodstream, significantly contributes to the burden of pneumococcal diseases, and is another major health concern in Germany [8]. It is a common cause of community-acquired pneumonia (CAP) and affects both children and adults, with higher incidence rates in the elderly and individuals with comorbidities [9]. Non-bacteremic pneumococcal pneumonia (NBPP) is associated with significant morbidity and mortality, particularly among high-risk groups, and leads to substantial healthcare utilization, including both outpatient and inpatient services [10].

The incidence of IPD in German children declined from 2006 to 2019 following the introduction of PCV7 in the NIP [7]. Vaccine-type PD also declined in adults over this period; however, the overall burden of PD in this population increased, particularly in older adults [7]. From 2020 to 2021, IPD incidence declined with the implementation of non-pharmaceutical interventions employed during the SARS-CoV-2 pandemic; however, since that time, IPD incidence in Germany has returned to or exceeded pre-pandemic levels [11].

Pneumococcal vaccination recommendations continue to be evaluated in Germany to address the growing burden of NBPP and IPD, especially among older adults. A 20-valent pneumococcal conjugate vaccine (PCV20) was approved in the European Union (EU) in 2022, and has been recommended as the only adult vaccine since September 2023 in Germany; in addition, an adult-focused 21-valent PCV (V116) received a positive opinion from the EU’s Committee for Medicinal Products for Human Use (CHMP) in January 2025, and has been approved by the EU commission in late March 2025 [12]. This analysis evaluated the impact of PCV20 and V116 as a replacement for PPSV23 on the epidemiological burden of pneumococcal disease among older adults in Germany, by comparing these two scenarios as well as assessing the implications of increasing vaccination uptake rates among this population, in combination with the re-vaccination of individuals previously vaccinated with PPSV23.

## 2. Methods

We adapted a previously published age-stratified S-I-S ordinary differential equation dynamic transmission model (DTM) of pneumococcal carriage transmission to the German setting, by updating demographic and epidemiological inputs [10,13] (see Appendix A). The DTM was stratified by age into six age strata: <2 years, 2–5 years, 5–18 years, 18–50 years, 50–60 years, and ≥60 years. Individuals in different age strata interacted based on mixing rates defined by an age-stratified contact matrix derived from published data for Germany [14]. The DTM was further stratified by grouping pneumococcal serotypes into 11 serotype classes (STCs), based on inclusion in various current, past, and future vaccines, including PCV7, PCV13, PCV15, PCV20, PCV21 (V116), and PPSV23 (Table 1). The classification is based on whether a serotype is targeted by an individual vaccine so that the assessment of each vaccine can be carried out by grouping some of the classes. For example, PPSV23 includes STCs 1–4 and STCs 6–9, while V116 includes STCs 3–10 with STC 10 representing the unique serotypes not in previous vaccines. Corresponding parameters were estimated according to the age and serotype grouping.

Concurrent cocolonization with serotypes from up to three STCs was possible, based on vaccination status and inter-serotype competition [10,13]. Progression to pneumococcal disease (i.e., IPD, inpatient NBPP, and outpatient NBPP) was estimated by multiplying the number of colonization events by the disease-specific case-to-carrier ratio. The model estimated pneumococcal disease burden in Germany under varying future vaccination strategies over a 10-year time-horizon.

We estimated age- and serotype-specific transmission and competition coefficients (for competition between colonized and invading serotypes), disease-specific case-to-carrier ratios, and vaccine efficacies against carriage acquisition through model calibration to German historical time series data for the incidence of IPD and NBPP, and to regional (UK) pre-vaccine pneumococcal carriage data [15,16,17,18]. The calibration process involved multiple steps. First, the model was calibrated to pre-vaccine pneumococcal carriage and IPD data, followed by another calibration from the onset of the first vaccine introduction to the currently recommended vaccine (while ensuring that all vaccine policies through the years were fully captured). Once calibration to IPD incidence data was achieved, a secondary calibration was conducted to estimate the case-to-carrier ratios of NBPP (inpatient and outpatient) by fitting the model to NBPP data, respectively. Calibrated parameters and model fitting results are shown in Appendix A. The model calibration was performed by minimizing the weighted sum of squared errors over annual age- and serotype-specific data utilizing the NMinimize function in Mathematica 14.1.

The target VCR for pediatric vaccination was assumed to remain constant at 85% over the entire time horizon, with an even split of PCV13 and PCV15 among children under two years old under the 2 + 1 vaccination schedule. Vaccine effectiveness (VE) against IPD and NBPP for pediatric vaccination varied by serotype and was based on published data [19,20,21,22,23]. For adult vaccination, VE against IPD for serotypes included in PCV13 was assumed to extend to covered serotypes in the V116 and PCV20 vaccines. VE estimates for covered serotypes were set to 75%, except for serotype 3, which was estimated at 26% [19,24]. For pediatric vaccination, VE against IPD for PPSV23 varied by serotype, based on published data [25,26]. Age-group specific VEs against IPD and NBPP are shown in Appendix A. The average duration of protection for all PCV vaccines was assumed to be 10 years; the average duration of protection for PPSV23 was assumed to be 7.5 years [27,28]. Germany-specific input parameters are summarized in Table 2.

We evaluated the impact of three vaccines (PPSV23, V116, and PCV20) on the adult population aged 18+ years in Germany, while maintaining the status quo pediatric vaccination program across all vaccination scenarios and over the entire time horizon. For the reference scenario (scenario 0), we assumed that PPSV23 would continue at the current VCR level (30%). For all other scenarios, we assumed that PPSV23 would be replaced by either PCV20 or V116 for all adult vaccination moving forward. We varied adult vaccination along three dimensions (Table 3):Vaccine formulation—replacement of PPSV23 (scenario 0) with either PCV20 (scenarios 1–4) or V116 (scenarios 5–8);Vaccine uptake—continuing at 30% uptake (scenarios 1, 2, 5, and 6), or increasing to 60% uptake (scenarios 3, 4, 7, and 8);Revaccination of adults previously vaccinated with PPSV23—no revaccination (scenarios 1, 3, 5, and 7), or revaccination at 50% uptake with the same vaccine formulation as for routine vaccination (scenarios 2, 4, 6, and 8).

## 3. Results

### 3.1. Invasive Pneumococcal Disease

The model projected an increase in IPD incidence in adults 18+ years over the 10-year time horizon under all vaccination scenarios, with the greatest increase resulting from the continuation of PPSV23 at the uptake of 30% (scenario 0) (Figure 1). Following an initial decline in incidence from the 2020 incidence of 5.83 IPD cases per 100,000 population, PCV20 with 30% uptake (scenario 1) led to a subsequent rebound in IPD which exceeded the 2020 level within five years; with V116 with 30% uptake (scenario 5), this return in IPD was delayed by an additional four years (Figure 1). Increasing uptake (scenarios 3 and 7) eliminated the return in IPD. Replacing PPSV23 with PCV20 (scenario 1) or V116 (scenario 5) at 30% uptake led to 4% or 8% lower overall IPD incidence at the 10-year time horizon, respectively, when compared with maintaining PPSV23 at 30% uptake (i.e., reference strategy) (Appendix A). IPD incidence was 4–36% lower for all vaccination strategies for all age groups when compared with the reference strategy, with the greatest reduction resulting from scenario 8 (vaccination with V116 with 60% uptake and 50% revaccination) (Appendix A).

Scenarios implementing V116 led to greater reductions in IPD incidence than equivalent scenarios implementing PCV20. At both 30% and 60% uptake, and with or without revaccination, the introduction of V116 resulted in fewer IPD cases among adults 18+ years than the introduction of PCV20, when compared with the reference strategy (scenario 0; PPSV23, 30% uptake, no revaccination) (Table 4). Vaccination with V116 led to 3.8–8.8% lower overall IPD annual incidence among adults aged 18+ years and 6.2–16.1% lower for adults aged 60+ years, at the 10-year time horizon when compared with equivalent scenarios utilizing PCV20, resulting in 5.3–11.3% fewer cumulative cases over the 10-year time horizon for adults 60+ years (Appendix A). Because V116 averted more cases than PCV20 in adults aged 60+, the proportion of cumulative IPD cases associated with 60+ was lower in V116 strategies than in PCV20 strategies (Table 5).

As expected, for V116 scenarios, there were more cases of IPD caused by serotypes covered by PCV20 but not by V116 than there were under the equivalent PCV20 scenario. Conversely, for PCV20 scenarios, there were more cases of IPD caused by serotypes covered by V116 but not by PCV20 than there were under the equivalent V116 scenarios (Appendix A). Nevertheless, V116 scenarios resulted in fewer cumulative IPD cases overall and for all adults 18+ years.

Increasing uptake to 60% reduced overall IPD incidence among adults 18+ years by 7.0–12.4% and incidence in adults 60+ years by 11.5–21.8%, leading to a 5.4–9.9% reduction in cumulative cases in this age group when compared to equivalent scenarios with 30% uptake (Appendix A). The revaccination of individuals previously vaccinated with PPSV23 led to 5.4–9.9% fewer cumulative IPD cases in adults 60+ over the time horizon. The relative performance of V116 versus PCV20 was not impacted by either increasing VCR and/or revaccination, i.e., overall IPD cases were consistently lower for V116 scenarios than PCV20 (Appendix A).

### 3.2. Non-Bacteremic Pneumococcal Pneumonia—Inpatient Cases

As observed with IPD incidence, the model projected an increase in inpatient NBPP incidence among adults aged 18+ years over the 10-year time horizon under all vaccination scenarios, with the greatest increase from the continued use of PPSV23 at 30% VCR (scenario 0) (Figure 2). As with IPD, inpatient NBPP incidence declined initially from the 2020 level of 269.4 NBPP inpatient cases per 100,0000 population, followed by an increase above 2020 levels within four years for PCV20 at 30% uptake (scenario 1); for V116 at 30% uptake (scenario 5) inpatient NBPP incidence had not yet returned to 2020 levels by the end of the 10-year time horizon. Increasing uptake with PCV20 or V116 (scenarios 3 and 7, respectively) prevented a rebound in incidence. Replacing PPSV23 with PCV20 with 30% uptake (scenario 1) led to 4% lower overall inpatient NBPP incidence at the 10-year time horizon, with 5% lower incidence in adults 60+ years, when compared with maintaining the reference strategy. In contrast, replacement with V116 (scenario 5) led to 10% lower overall inpatient NBPP incidence, 6% and 5% lower in 18–50- and 50–60-year-olds, respectively, and 14% lower in adults 60+. Inpatient NBPP incidence was 4–40% lower than the reference strategy for all vaccination strategies for all age groups, with the lowest incidence occurring in scenario 8 (vaccination with V116 with 60% uptake and 50% revaccination) (Appendix A).

As with IPD, scenarios that implemented V116 led to lower inpatient NBPP incidence than equivalent scenarios implementing PCV20. As with cumulative adult IPD cases, at both 30% and 60% uptake, both with and without revaccination, the implementation of V116 prevented a greater number of inpatient NBPP cases among adults than the implementation of PCV20 when compared with the reference strategy (scenario 0) (Table 4). Vaccination with V116 led to 6.9–16.3% lower overall inpatient NBPP incidence, 10.1–25.3% lower for adults aged 60+ years, at the 10-year time horizon when compared with equivalent scenarios utilizing PCV20, resulting in 8.9–18.4% fewer cumulative inpatient NBPP cases over the time horizon for adults aged 60+ years (Appendix A). As seen with IPD, because V116 averted more inpatient NBPP cases than PCV20 in adults aged 60+, the proportion of cumulative inpatient NBPP cases associated with 60+ was lower in V116 strategies than in PCV20 strategies (Table 6).

As expected, and as observed with IPD cases, for V116 scenarios, there were more inpatient NBPP cases caused by serotypes covered by PCV20 but not V116, than there were under the equivalent PCV20 scenarios. Conversely, for PCV20 scenarios, there were more inpatient NBPP cases caused by serotypes covered by V116 but not PCV20, than there were under the equivalent V116 scenarios (Appendix A). Nevertheless, V116 scenarios resulted in fewer cumulative inpatient NBPP cases (independent of serotype) overall and for all adults aged 18+ years.

Increasing VCR from 30% to 60% led to 6.5–15.4% lower overall inpatient NBPP incidence 9.4–23.9% lower in adults aged 60+ years, leading to 5.5–12.9% fewer cumulative cases in this age group (Appendix A). The revaccination of individuals previously vaccinated with PPSV23 led to 4.3–10.7% fewer cumulative inpatient NBPP cases in adults aged 60+ over the time horizon. As seen with IPD cases, the relative performance of V116 versus PCV20 was not impacted by either increasing VCR and/or revaccination, i.e., overall inpatient NBPP cases were consistently lower for V116 scenarios than PCV20. (Appendix A).

### 3.3. Non-Bacteremic Pneumococcal Pneumonia—Outpatient Cases

As above, the model projected an increase in overall outpatient NBPP incidence in adults aged 18+ years over the 10-year time horizon under all vaccination scenarios, with the greatest increase resulting from the continuation of PPSV23 at 30% VCR (scenario 0) (Figure 3). For PCV20 (scenario 1), outpatient NBPP incidence remained above the 2020 level 262.3 NBPP outpatient cases per 100,000 population for the entire time horizon; for V116 (scenario 5), incidence dropped initially, then increased, surpassing 2020 levels five years after introduction. Increasing uptake to 60% for PCV20 (scenario 3) delayed this rebound to 2023, whereas increasing uptake for V116 (scenario 7) prevented the rebound for the entirety of the time horizon. Replacing PPSV23 with PCV20 (scenario 1) led to a 3% lower overall outpatient NBPP incidence at the 10-year time horizon, 5% lower in adults aged 60+ years, when compared with the reference strategy. In contrast, replacement with V116 (scenario 5) led to 7% lower overall outpatient NBPP incidence, 6% and 5% lower in 18–50- and 50–60-year-olds, respectively, and 14% lower in adults aged 60+ years (Appendix A).

As with cumulative adult IPD cases and cumulative adult inpatient NBPP cases, at both 30% and 60% VCR, and both with and without revaccination, the implementation of V116 prevented a greater number of outpatient NBPP cases among adults over the 10-year time horizon than did the implementation of PCV20, when compared with the reference strategy (scenario 0) (Table 4).

Vaccination with V116 led to 4.1–9.0% lower overall outpatient NBPP incidence, 10.1–25.3% lower for adults aged 60+ years, at the 10-year time horizon when compared with equivalent scenarios utilizing PCV20, resulting in 8.9–18.4% fewer cumulative outpatient NBPP cases over the time horizon for adults aged 60+ years (Appendix A). As seen with IPD and inpatient NBPP, because V116 averted more outpatient NBPP cases than PCV20 in adults aged 60+ years, the proportion of cumulative outpatient NBPP cases associated with adults aged 60+ years was lower in V116 strategies than in PCV20 strategies (Table 7).

As expected, and as observed for both IPD and inpatient NBPP cases, for V116 scenarios, there were more outpatient cases of NBPP caused by serotypes covered by PCV20, but not V116, than there were under the equivalent PCV20 scenarios. Conversely, for PCV20 scenarios, there were more outpatient cases of NBPP caused by serotypes covered by V116 but not PCV20, than there were under the equivalent V116 scenarios (Appendix A). Nevertheless, V116 scenarios resulted in fewer cumulative outpatient NBPP cases (independent of serotype) overall and for all adults aged 18+ years.

Increasing the uptake from 30% to 60% reduced the overall outpatient NBPP incidence in adults aged 18+ years by 4.0–8.6% and the incidence in adults aged 60+ years by 9.4–23.9%, leading to a 5.5–12.9% reduction in cumulative cases in this age group (Appendix A). The revaccination of individuals previously vaccinated with PPSV23 led to 4.3–10.7% fewer cumulative outpatient NBPP cases in adults aged 60+ years over the time horizon. As seen with IPD and inpatient NBPP cases, the relative performance of V116 versus PCV20 was not impacted by either increasing VCR and/or revaccination, i.e., overall outpatient NBPP cases were consistently lower for V116 scenarios than PCV20, with greater reductions resulting from vaccination with V116 (Appendix A).

## 4. Discussion

At present, the uptake of pneumococcal vaccine among adults aged 60+ years is low, with approximately 30% of older adults having received PPSV23 vaccination. With the availability of multiple adult pneumococcal vaccines in Germany, there is an opportunity to assess the adult vaccination program by evaluating different vaccines, coverage rates, and revaccination. In this analysis, we adapted a dynamic model for the transmission of *S. pneumoniae* carriage to the German setting and evaluated eight potential adult vaccination strategies varying vaccine type, uptake rate, and revaccination rate [10,13].

Pneumococcal disease due to IPD and NBPP was projected to increase in all adults aged 18+ years in Germany over a 10-year time horizon under the continuation of PPSV23. When PPSV23 was replaced by either V116 or PCV20, IPD and NBPP inpatient and outpatient incidence in this age group were projected to increase at a significantly slower rate than under the reference strategy of continuation of PPSV23. IPD and NBPP incidence among adults 60+ years were projected to be lower at the 10-year time horizon with the implementation of V116 versus PCV20.

Increasing VCR from 30% to 60% was projected to lead to a noticeable reduction in pneumococcal disease over the time horizon. Scenarios with 60% uptake (scenarios 3 and 7) were associated with fewer cumulative IPD and NBPP cases in the overall German population compared with 30% uptake scenarios, and noticeably fewer cases in the target population of adults 60+ years. These clinical improvements were greater when V116 was implemented, rather than PCV20.

The greatest projected overall impact on clinical outcomes resulted from the simultaneous implementation of increasing uptake from 30% to 60% and revaccinating previously PPSV23-vaccinated individuals (scenarios 4 and 8). The combination of increased uptake and revaccination was projected to lead to the lowest cumulative IPD and inpatient and outpatient NBPP cases over all age groups by the end of the 10-year time horizon. Scenario 8, which included V116 adult vaccination and V116 revaccination of PPSV23-vaccinated individuals, led to the greatest improvements in clinical outcomes across all scenarios.

The model projected that increasing adult VCRs and using V116 versus PCV20 would noticeably reduce the incidence of IPD and NBPP in Germany, and that revaccinating individuals previously vaccinated with PPSV23 would further reduce pneumococcal disease burden.

This analysis was subject to several limitations. First, the DTM assumed a static population which did not account for changes in the population age distribution over time. Given Germany’s aging population, this assumption may lead to an underestimation of the impacts of varying vaccination strategies among older adults [33]. In addition, due to the grouping of serotypes into STCs, the DTM is not capable of disaggregating single serotypes for discussion and prevents a detailed analysis of serotypes which may lead to higher disease in specific subpopulations. The incorporation of this capability into future versions of the DTM is planned. Further, non-pharmaceutical interventions implemented in Germany during the SARS-CoV-2 pandemic, which had a noticeable influence on pneumococcal disease, were not accounted for in this model, as the most recent year in the calibration dataset was 2019. Nevertheless, published data show that pneumococcal incidence rapidly returned to pre-pandemic levels following the cessation of these interventions [11].

Another limitation of this analysis lies in the use of UK-specific pneumococcal carriage data as a proxy for Germany-specific pre-PCV carriage data. Due to the unavailability of Germany-specific pneumococcal carriage data, reliance on UK data was necessary to complete the model calibration and derive meaningful insights from the model projections. However, differences in demographic, epidemiological, environmental, and healthcare factors between Germany and the UK could influence pneumococcal carriage patterns, rendering the proxy an imperfect representation of the German population. These factors may limit the generalizability of the findings to Germany-specific contexts and should be taken into consideration when interpreting the results. Future studies would benefit from concerted efforts to collect and analyze Germany-specific pneumococcal carriage data to improve accuracy and contextual relevance. Despite these limitations, this study serves as an important stepping-stone in addressing the broader research question and highlights areas requiring further investigation.

Finally, the historical IPD and NBPP incidence data utilized in model calibration were collected over a long time period, and included variations in reporting rate over time, which in turn may impact model outcomes.

## 5. Conclusions

Though the uptake of pneumococcal vaccine among adults 60+ years in Germany is currently low, the vaccination of older adults with V116 and implementing strategies to increase vaccine uptake and revaccinate previously vaccinated individuals was predicted to lead to substantial reductions in pneumococcal disease burden over a 10-year time horizon. Policymakers should consider taking advantage of the opportunity to reevaluate the adult vaccination program in alignment with these strategies to reduce the disease burden of *S. pneumoniae* in Germany.

## Figures and Tables

**Figure 1 vaccines-13-00475-f001:**
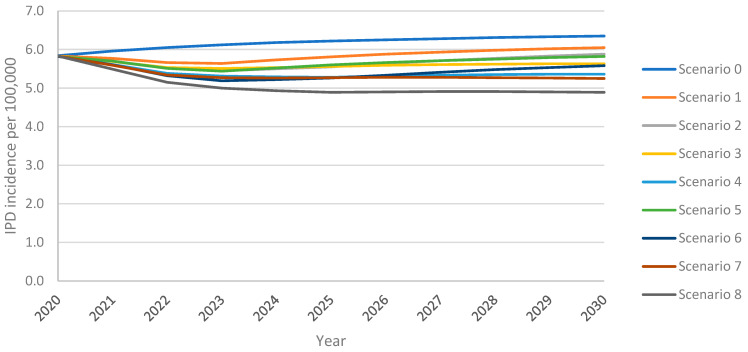
Overall IPD incidence among adults 18+ years over the 10-year time horizon under varying vaccination scenarios.

**Figure 2 vaccines-13-00475-f002:**
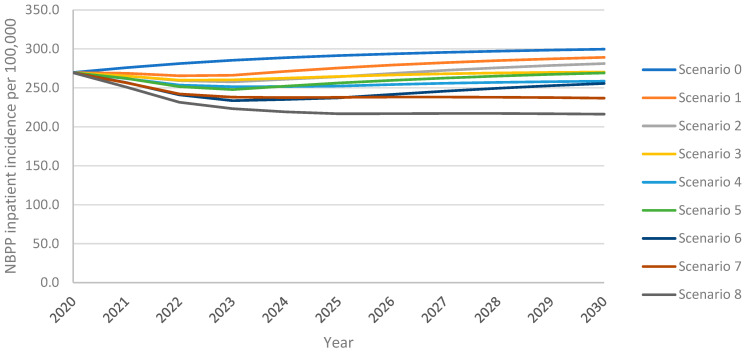
Overall inpatient NBPP incidence in adults aged 18+ years over the 10-year time horizon under varying vaccination scenarios.

**Figure 3 vaccines-13-00475-f003:**
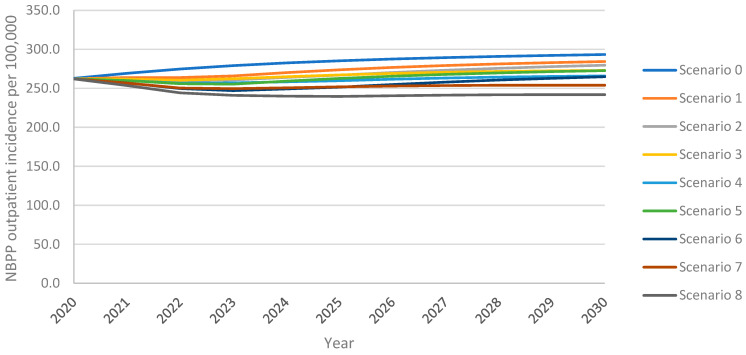
Overall outpatient NBPP incidence in adults aged 18+ years over the 10-year time horizon under varying vaccination scenarios.

**Table 1 vaccines-13-00475-t001:** Grouping of pneumococcal serotypes into serotype classes, and visualization of inclusion of classes in vaccines included in the current and historical German NIP. ST = serotype; STC = serotype class.

Serotype Class	Serotypes Included in Class	Description	PCV7	PCV13	PCV15	PCV20	V116	PPSV23
1	4, 6B, 9V, 14, 18C, 19F, 23F	PCV7 ST						
2	1, 5	PCV13 ST minus 3, 6A, 7F, & 19A						
3	3	ST3						
4	7F, 19A	Additional PCV13 ST						
5	6A, 6C	Additional PCV13 ST + 6C cross protection						
6	22F, 33F	Additional PCV15 ST						
7	9N, 17F, 20	Additional PPSV23 ST includes STC1–4, 6–9						
8	8, 10A, 11A, 12F	Additional PCV20 ST includes STC1–7, 10						
9	15B	Additional PCV20 ST						
10	15A, 15C, 16F, 23A, 23B, 24F, 31, 35B	Additional V116 ST excludes STC1–2, 10						
11	(NVTs)	NVTs						

Colored cells represent inclusion of the given serotype class in the vaccine.

**Table 2 vaccines-13-00475-t002:** Germany-specific input parameter values for the demographic and transmission model components of the pneumococcal dynamic transmission model.

Parameter	Value	Source
**Demographic model component**		
Fertility rate	Varies	[29]
Mortality rate	Varies	[30]
Aging rate	Varies	Calculated
Contact mixing rate	Varies	[14]
**Transmission model component**		
**Case fatality rates**		
IPD (all serotype groups)	0–5 years: 1.46% 5–18 years: 1.68% 18–50 years: 4.62% 50–60 years: 12.48% 60+ years: 15.73%	[31]
NBPP (deaths per 100 cases)	0–16 years: 0 16–50 years: 4.42 50–60 years: 10.95 60–70 years: 14.10 70+ years: 20.36	0–16 years: assumption 16+ years: [1]
**NBPP case distribution**		
Inpatient	0–16 years: 41.4% 16–50 years: 20.9% 50–60 years: 34.2% 60+ years: 64.7%	[32]
Outpatient	0–16 years: 58.6% 16–50 years: 79.1% 50–60 years: 65.8% 60+ years: 35.3%	[32]
**Historic vaccination coverage**		
Pediatric PCV (ages 0–2 years)	2006: 20% 2007: 50% 2008: 67% 2009: 69% 2010: 71% 2011: 71% 2012: 72% 2013: 71% 2014: 70% 2015: 68% 2016: 70% 2017: 71% 2018: 73% 2019: 75%	[5]
Adult PPSV23 (ages 60+ years)	1998–2015: 22% 2016: 24% 2017: 26% 2018: 28% 2019: 30%	[5]
**Vaccine effectiveness**		
PCV vaccines (ages 60+ years)	Serotype 3: 26% All other covered serotypes: 75%	[19]
PPSV23 (ages 60+ years)	Varies by serotype	[25]
**Vaccine duration of protection**		
PCV vaccines	10 years	[27]
PPSV23	7.5 years	[28]

IPD: invasive pneumococcal disease; NBPP: non-bacteremic pneumococcal pneumonia.

**Table 3 vaccines-13-00475-t003:** Adult vaccination scenarios evaluated over a 10-year time horizon.

Scenario	Vaccine	Uptake	Revaccination
Scenario 0	PPSV23	30%	-
Scenario 1	PCV20	30%	-
Scenario 2	PCV20	30%	50%
Scenario 3	PCV20	60%	-
Scenario 4	PCV20	60%	50%
Scenario 5	V116	30%	-
Scenario 6	V116	30%	50%
Scenario 7	V116	60%	-
Scenario 8	V116	60%	50%

**Table 4 vaccines-13-00475-t004:** Impact of adult vaccination scenarios on incremental cumulative clinical outcomes in adults 18+ years over 10-year time horizon.

Clinical Outcome	Without Revaccination		With 50% Revaccination	
	PCV20	V116	PCV20	V116
30% VCR				
Averted IPD cases	2856	4388	4404	6433
Averted NBPP inpatient cases	110,338	247,963	178,240	362,519
Averted NBPP outpatient cases	84,116	163,808	125,244	230,647
60% VCR				
Averted IPD cases	4820	6996	6631	9410
Averted NBPP inpatient cases	196,688	394,477	276,063	530,681
Averted NBPP outpatient cases	136,264	249,109	184,334	328,491

**Table 5 vaccines-13-00475-t005:** Impact of adult vaccination scenarios on cumulative IPD cases over 10-year time horizon (total cases).

Age Group	Without Revaccination			With Revaccination	
	PPSV23	PCV20	V116	PCV20	V116
30% VCR					
18–50	5846	5632	5596	5597	5558
50–60	7660	7416	7371	7361	7311
60+	29,587	27,189	25,738	25,731	23,791
Total adult	43,093	40,237	38,705	38,689	36,660
60% VCR					
18–50	N/A	5589	5549	5549	5505
50–60		7349	7296	7286	7226
60+		25,335	23,252	23,627	20,952
Total adult		38,273	36,097	36,462	33,683

**Table 6 vaccines-13-00475-t006:** Impact of adult vaccination scenarios on cumulative inpatient NBPP cases over 10-year time horizon (total cases).

Age Group	Without Revaccination			With Revaccination	
	PPSV23	PCV20	V116	PCV20	V116
30% VCR					
18–50	131,842	126,244	125,156	125,374	124,239
50–60	121,560	117,393	116,618	116,483	115,636
60+	1,632,982	1,532,409	1,396,647	1,466,287	1,283,990
Total adult	1,886,384	1,776,046	1,638,421	1,708,144	1,523,865
60% VCR					
18–50	N/A	125,171	124,023	124,154	122,951
50–60		116,267	115,399	115,209	114,250
60+		1,448,258	1,252,485	1,370,958	1,118,502
Total adult		1,689,696	1,491,907	1,610,321	1,355,703

**Table 7 vaccines-13-00475-t007:** Impact of adult vaccination scenarios on cumulative outpatient NBPP cases over 10-year time horizon (total cases).

Age Group	Without Revaccination			With Revaccination	
	PPSV23	PCV20	V116	PCV20	V116
30% VCR					
18–50	500,020	478,789	474,663	475,488	471,183
50–60	233,552	225,545	224,057	223,797	222,170
60+	891,036	836,158	762,080	800,079	700,608
Total adult	1,624,608	1,540,492	1,460,800	1,499,364	1,393,961
60% VCR					
18–50	N/A	474,719	470,366	470,860	466,299
50–60		223,384	221,715	221,351	219,508
60+		790,241	683,418	748,063	610,310
Total adult		1,488,344	1,375,499	1,440,274	1,296,117

## Data Availability

All relevant data are within the manuscript and its Appendix A files. This is a modeling study and, therefore, no primary data were collected in this study. All inputs were from published literature and included only anonymized data.

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
