# Peer review of "Epidemiological Impact of Increasing Vaccination Coverage Rate and Re-Vaccination on Pneumococcal Disease in Older Adults in Germany"

_vaccines, 2025, doi:10.3390/vaccines13050475_

Round 1
Reviewer 1 Report
Comments and Suggestions for Authors
The manuscript is well-written, and I did not find any significant flaws with its methodology or conclusions. However, its presentation will benefit from:
- following the ISPOR-SMDM guidance on modelling studies for health economics and outcome research (https://doi.org/10.1016/j.jval.2012.06.012)
- uploading the CHERRS 2022 checklist.
Author Response
The manuscript is well-written, and I did not find any significant flaws with its methodology or conclusions. However, its presentation will benefit from:
- following the ISPOR-SMDM guidance on modelling studies for health economics and outcome research (https://doi.org/10.1016/j.jval.2012.06.012)
- uploading the CHERRS 2022 checklist.
Response: Thank you for your comments. We have confirmed that our manuscript follows the ISPOR-SMDM guidance for dynamic transmission models. Thank you also for the referral to the CHEERS 2022 checklist. We have endeavored to follow this checklist in the drafting of our manuscript, however not all categories apply as this is an epidemiological analysis as opposed to an economic evaluation.
Reviewer 2 Report
Comments and Suggestions for Authors
The authors from MSD, vaccine V116 manufacturer, present the study on epidemiological impact of increasing pneumococcal vaccine coverage in older adults aged 60+ years and at risk adults in Germany.They evaluated three adult vaccination scenarios over 10 years using PPS23, PCV20 and V116 with 30% and 60% uptake. The aim of the study was to show the impact of pneumococcal vaccination on the incidence of invasive pneumococcal diseases (IPD) and non-bacteremic pneumococcal pneumonia (NBPP) in out-and in-patients.The dynamic transmission model was used.
In their model the authors found with 30% and 60% VCR at the 10- year time higher impact on the incidence of IPD and NBPP (in-and out-patients) with V116 than with PCV20.
My comments:
-statistical analysis is not done
-the serotype dinamics over 10 years period was not mentioned
-serotype 4 is rare in children but higher in adults and was not discussed. It is not covered with V116.
-on Figure 1 and on Figure 4 on y axis only the % are written and not absolute numbers at the start of evaluation. Can be mentioned in the text. It would be fine for readers to have such numbers to see where we start.
-the paper should be reviewed by statistician and expert for the methodology
-is the vaccine V116 already approved by EMA?
Author Response
My comments:
-statistical analysis is not done
Response: Thank you for your comment. While statistical analyses can be insightful in certain contexts, this paper would not benefit from performing additional statistical analysis. In this paper, we are quantifying the impact of new vaccines in comparison to a current vaccine via different scenario analyses. Incorporating statistical analysis could potentially shift the focus away from the primary purpose of the paper. All vaccines were treated equally so any variations in any parameters would affect all the vaccines equally and not expected to change the conclusions of the study.
-the serotype dynamics over 10 years period was not mentioned
Response: Thank you for your comment. The manuscript’s focus is on employing multiple scenarios to quantify the impact of the different vaccines, making the discussion around serotype dynamics impossible for all 9 scenarios. However, a future analysis which focuses on using a single scenario is planned, to discuss serotype dynamics from using the vaccines beyond the time horizon.
-serotype 4 is rare in children but higher in adults and was not discussed. It is not covered with V116.
Response: Thank you for pointing this out. We have added clarifying text to the Methods section (lines 106-112 in the tracked changes version of manuscript), to explain how serotypes in the model are grouped together into serotype classes based on inclusion in various vaccines. While this simplification reduces the computational requirements of the model, as a result the model is not capable of disaggregating a single serotype for discussion. Additional text to this effect has been added to the discussion (lines 351-355 in the tracked changes version of the manuscript).
-on Figure 1 and on Figure 4 on y axis only the % are written and not absolute numbers at the start of evaluation. Can be mentioned in the text. It would be fine for readers to have such numbers to see where we start.
Response: Thank you for your suggestion. We have added the initial incidence values at the beginning of the time horizon to the text referring to Figures 1, 2, and 3 (lines 172, 219-220, and 269-270 in the tracked changes version of the manuscript). Please note there is no Figure 4.
-the paper should be reviewed by statistician and expert for the methodology
Response: Thank you for this recommendation. The model and the manuscript have both been reviewed by multiple experts from initial model development stages through to manuscript submission, and the methodology has been approved as correct. As this study was not a statistical analysis, review by a statistician is not appropriate.
-is the vaccine V116 already approved by EMA?
Response: Yes, V116 has been approved by the EMA. We have updated the referring sentence in the Introduction (line 92 in the tracked-changes version of the manuscript) and reference #12 accordingly.
Reviewer 3 Report
Comments and Suggestions for Authors
In this manuscript titled “Epidemiological Impact of Increasing Vaccination Coverage Rate and Re-Vaccination on Pneumococcal Disease in Older Adults in Germany”, authors evaluated the impact of pneumococcal vaccination strategies (PCV20 and V116) on disease burden in older German adults using a dynamic transmission model. The results confirmed that vaccination with PCV20 and V116 initially decreased PD incidence, but incidence returned to pre-vaccine levels after five and eight years, respectively. However, there exists many minor problems in this manuscript, which need further revision and improvement. The specific amendments are as follows:
Major issues:
- The model relies on UK pneumococcal carriage data as a proxy for pre-vaccine conditions in Germany, but the justification for this choice is insufficient. Provide evidence supporting the validity of UK data for the German context (e.g., similarity in serotype distribution, healthcare practices). If German-specific carriage data are unavailable, explicitly acknowledge this limitation and discuss its potential impact.
- Table 1: Provide a clearer rationale for serotype class (STC) groupings, especially for V116-specific STCs (Class 10).
Minor issues:
- Abstract: Replace repetitive "rebound" with synonyms (e.g., "resurgence").
- In line 104, “see Table 1” should be deleted “see”.
- Table 5 (60% VCR) includes "N/A" for the 18-50, 50-60, and 60+ age groups under PCV20. Clarify whether this is a data omission or intentional exclusion.
- In line 75, define "NBPP" at first mention.
- Specify whether "revaccination" applies only to PPSV23-vaccinated individuals or includes naturally immune populations.
- Reference 13 (Malik et al., 2024) is cited as a preprint. Update with journal details if now published.
Author Response
Major Issues:
- The model relies on UK pneumococcal carriage data as a proxy for pre-vaccine conditions in Germany, but the justification for this choice is insufficient. Provide evidence supporting the validity of UK data for the German context (e.g., similarity in serotype distribution, healthcare practices). If German-specific carriage data are unavailable, explicitly acknowledge this limitation and discuss its potential impact.
Response: Thank you for pointing this out. We have included a detailed discussion on the issues regarding the use of UK pneumococcal carriage data as proxy for pre-vaccine conditions in Germany in the Discussion section (lines 361-373 in the tracked-changes version of the manuscript), mentioning the unavailability of Germany-specific carriage data and the consequence of using UK carriage data as proxy.
- Table 1: Provide a clearer rationale for serotype class (STC) groupings, especially for V116-specific STCs (Class 10).
Response: Thank you for your recommendation. We have included a detailed description of the rationale for the STC groupings in the Method section (lines 106-113 in the tracked-changes version of the manuscript).
Minor Issues:
- Abstract: Replace repetitive "rebound" with synonyms (e.g., "resurgence").
Response: Thank you for your suggestion. We have replaced the word “rebound” with “resurgence” in the abstract (line 23 in the tracked-changes version of the manuscript).
- In line 104, “see Table 1” should be deleted “see”.
Response: Thank you for your recommendation. We have deleted the word “see” from “see Table 1” (line 108 in the tracked-changes version of the manuscript).
- Table 5 (60% VCR) includes "N/A" for the 18-50, 50-60, and 60+ age groups under PCV20. Clarify whether this is a data omission or intentional exclusion.
Response: Thank you for your comment. Please note that “N/A” in Table 5 is under PPSV23 not PCV20. This was intentionally omitted as PPSV23 is expected to be discontinued in the future.
- In line 75, define "NBPP" at first mention.
Response: Thank you for pointing this out. The acronym “NBPP” is now defined at its first use (lines 75-76 in the tracked-changes version of the manuscript).
- Specify whether "revaccination" applies only to PPSV23-vaccinated individuals or includes naturally immune populations.
Response: Thank you for your comment. As noted in lines 165-167 (tracked-changes version of the manuscript), revaccination applies only to adults previously vaccinated with PPSV23.
- Reference 13 (Malik et al., 2024) is cited as a preprint. Update with journal details if now published.
Response: Thank you for pointing this out. The paper by Malik et al has now been published, and reference #10 has been updated accordingly. Please note that reference #13 refers to a different study.
Round 2
Reviewer 2 Report
Comments and Suggestions for Authors
The authors included some changes in the text and answered to my comments.